

# Cloud top heights and aerosol layer properties from EarthCARE lidar observations: the A-CTH and A-ALD products

Ulla Wandinger[1], Moritz Haarig[1], Holger Baars[1], David Donovan[2], and Gerd-Jan van Zadelhoff[2]

[1]Leibniz Institute for Tropospheric Research, Leipzig, Germany
[2]Royal Netherlands Meteorological Institute, De Bilt, The Netherlands

**Correspondence:** Ulla Wandinger (ulla.wandinger@tropos.de)

**Abstract.**

The Atmospheric Lidar (ATLID) on the Earth Clouds, Aerosols and Radiation Explorer (EarthCARE) provides vertically resolved information on aerosols and clouds at the global scale. This paper describes the algorithms for the determination of cloud-top-height and aerosol-layer information from ATLID Level 1b (L1b) and Level 2a (L2a) input data. The ATLID L2a

Cloud Top Height (A-CTH) and ATLID Aerosol Layer Descriptor (A-ALD) products are developed to ensure the provision of atmospheric layer products in continuation of the heritage from the Cloud-Aerosol Lidar and Infrared Pathfinder Satellite Observations (CALIPSO). Moreover, the products serve as input for synergistic algorithms that make use of data from ATLID and EarthCARE's Multi Spectral Imager (MSI). Therefore, the products are provided on the EarthCARE Joint Standard Grid (JSG). A wavelet covariance transform (WCT) method with flexible thresholds is applied to determine layer boundaries from

the ATLID Mie co-polar signal. Strong features detected with a horizontal resolution of one JSG pixel (approximately 1 km) or 11 JSG pixels are classified as thick or thin clouds, respectively. The top height of the uppermost cloud layer together with information on cloud layering is stored in the A-CTH product for further use in the generation of the ATLID-MSI Cloud Top Height (AM-CTH) synergy product. Aerosol layers are detected as weaker features at a resolution of 11 JSG pixels. Layer-mean optical properties are calculated from the ATLID L2a Extinction, Backscatter and Depolarization (A-EBD) product and

stored in the A-ALD product, which also contains the aerosol optical thickness (AOT) of each layer, the stratospheric AOT, and the AOT of the entire atmospheric column. The latter parameter is used to produce the synergistic ATLID-MSI Aerosol Column Descriptor (AM-ACD) later in the processing chain. Several quality criteria are applied in the generation of A-CTH and A-ALD, and respective information is stored in the products. The functionality and performance of the algorithm are demonstrated by applying it to common EarthCARE test scenes. Conclusions are drawn for the application to real-world data

and the validation of the products after the launch of EarthCARE.

## 1 Introduction

The Earth Clouds, Aerosols and Radiation Explorer (EarthCARE) developed by the European Space Agency (ESA) and the Japan Aerospace Exploration Agency (JAXA) carries four sensors on one platform, a cloud-profiling radar (CPR), a high-spectral-resolution cloud/aerosol lidar (ATLID), a cloud/aerosol multi-spectral imager (MSI), and a three-view broad-band



radiometer (BBR) (Illingworth et al., 2015; Wehr et al., 2023). With the highly synergistic approach, the mission aims at un-
precedented accuracy in the observation of aerosols and clouds and their impact on the global radiation budget. The EarthCARE
mission requirements are based upon the need to derive the radiative flux at the top of the atmosphere (TOA) with an accuracy
of 10 Wm$^{-2}$ for a 100 km$^2$ snapshot view of the atmosphere. Accordingly, highly resolved information on the presence and
properties of aerosol and cloud layers for respective three-dimensional (3D) scenes is needed. The active instruments ATLID

and CPR contribute with vertically resolved measurements for a two-dimensional (2D) atmospheric cross section along the
satellite track (e.g., van Zadelhoff et al., 2023; Donovan et al., 2023a; Kollias et al., 2022; Mroz et al., 2023; Irbah et al., 2022).
The passive imager MSI provides observations of columnar aerosol and cloud properties across a 150 km wide swath (Docter
et al., 2023; Hünerbein et al., 2022, 2023), which are used to extend the 2D cross sections from lidar and radar into the 3D
domain (Haarig et al., 2023; Qu et al., 2022a). With this approach, 3D radiation modeling and closure assessments become

possible, i.e., modeled TOA radiances and fluxes can be compared with those derived from BBR measurements (Cole et al.,
2022; Barker et al., 2022).

As a prerequisite for the synergistic algorithms, highly accurate information on cloud top height (CTH) and aerosol layering
along track is needed from the lidar observations. The EarthCARE mission requirements postulate an accuracy of 300 m for
the determination of CTH for both ice and water clouds. With respect to aerosols, a detection threshold of 0.05 for aerosol

optical thickness (AOT) and an accuracy of the vertical layering of 500 m for a horizontal resolution of 10 km have been
defined (Wehr et al., 2023). EarthCARE's 355-nm high-spectral-resolution lidar ATLID is designed to achieve these goals by
measuring atmospheric backscattering with a horizontal resolution of 285 m (two laser shots on-board accumulation) and a
vertical resolution of approximately 100 m up to 20 km and 500 m from 20 to 40 km height (Wehr et al., 2023). The atmospheric
return is split into a co-polar and a cross-polar component with respect to the linear polarization of the emitted laser beam. The

co-polar component is further separated into a molecular (Rayleigh) and a particulate (Mie) part. The geolocated, calibrated,
and fully corrected signals are available in the ATLID Level 1b (L1b) product A-NOM (Eisinger et al., 2023). From the
three independent signals, and by using the ATLID Level 2a (L2a) Feature Mask product (A-FM) as an additional input (van
Zadelhoff et al., 2023), profiles of particle extinction and backscatter coefficients, lidar ratio, and particle linear depolarization
ratio at 355 nm are derived along the track of the satellite and the detected targets are classified. The variables are provided

in the ATLID L2a products A-EBD (ATLID Extinction, Backscatter, Depolarization), A-AER (ATLID Aerosol), and A-TC
(ATLID Target Classification). These products maintain the native vertical resolution of the L1b signals, while the horizontal
resolution is adapted to the measurement conditions and depends on the detected features (clouds and aerosols), the actual
signal-to-noise ratio (SNR), and the applied algorithms (Donovan et al., 2023a).

In this paper, we describe the algorithms for determining the lidar stand-alone L2a products ATLID Cloud Top Height (A-

CTH) and ATLID Aerosol Layer Descriptor (A-ALD), which make use of A-NOM, A-EBD, and A-TC and, in turn, serve
as input for the synergistic algorithms to produce the ATLID-MSI Cloud Top Height (AM-CTH) and ATLID-MSI Aerosol
Column Descriptor (AM-ACD; see EarthCARE production model, Eisinger et al., 2023). The products are generated with the
so-called ATLID Layer Products processor (A-LAY) on the EarthCARE Joint Standard Grid (JSG, Eisinger et al., 2023). Layer
detection is based on a wavelet covariance transform (WCT) technique with thresholds, which is applied to the Mie co-polar



signal. The thresholds can be configured such that the algorithm is suited for aerosol as well as cloud layer identification. The cloud part of the algorithm focuses on the detection of upper cloud boundaries. The A-CTH product contains information on the top height of the uppermost cloud layer and on the occurrence of multiple layers, when the upper layer is semi-transparent and penetrated by the laser beam. This information is especially useful in the context of CTH retrievals with MSI, which are strongly biased by semi-transparent clouds (Hünerbein et al., 2023). Thus, information from A-CTH is used to perform

respective corrections to the MSI retrievals along and across track, which are provided in the synergistic AM-CTH product (Haarig et al., 2023). For aerosol layers, the upper and lower boundaries are given in the A-ALD product. For each detected layer, the mean optical properties (extinction and backscatter coefficient, lidar ratio, particle linear depolarization ratio) and the layer AOT are calculated from the respective profiles of A-EBD. Furthermore, the columnar AOT, the stratospheric AOT, and the sum of layer AOT are stored in the product. The A-ALD product is intended to continue the heritage of aerosol layer

information available from the Cloud-Aerosol Lidar and Infrared Pathfinder Satellite Observations (CALIPSO, Vaughan et al., 2009). It also supports aerosol typing efforts and synergistic aerosol algorithms. EarthCARE mission requirements call for the quantification of absorbing or non-absorbing aerosols from natural and anthropogenic sources. Information on the spectral AOT is of interest in this context (see, e.g., Wandinger et al., 2022) and can be gained by combining ATLID measurements at 355 nm with MSI observations at 670 nm (over land and ocean) and 865 nm (over ocean) available from the MSI AOT product

(M-AOT, Docter et al., 2023). Respective Ångström exponents, together with track-to-swath extrapolations, are provided in the AM-ACD product (Haarig et al., 2023).

The paper is organized as follows. Sect. 2 gives an overview on the A-CTH and A-ALD products and summarizes their contents. The A-LAY algorithm is described in Sect. 3. After an overview of the processor, the selection of the layer-detection method is discussed and the WCT algorithm is introduced, followed by a description of the variables, quality indicators, and

consistency parameters contained in the products. Sect. 4 presents the algorithm validation based on the common test scenes from the EarthCARE End-to-End Simulator (ECSIM, Donovan et al., 2023b; Qu et al., 2022b). Major findings are summarized in the Conclusion.

## 2    ATLID layer products

The A-CTH and A-ALD products belong to the ATLID L2a layer products defined in the ESA EarthCARE production model

and product list (Wehr et al., 2023; Eisinger et al., 2023). Since their generation requires input from ATLID L2a profile data, they are produced at the end of the ATLID L2a processing. The products are prerequisites for synergistic ATLID-MSI algorithms and are thus generated on the JSG along the satellite track, i.e., their horizontal resolution is determined by the radar footprint, if radar measurements are available, and fixed to 1 km otherwise. The vertical resolution of the JSG corresponds to the native ATLID resolution, which therefore is maintained in the products.





## 2.1 ATLID Cloud Top Height (A-CTH)

The major variable contained in the A-CTH product is the top height of the uppermost cloud layer. This information is complemented with a simplified classification of the uppermost cloud, indicating thick, thin, and multi-layer clouds. The CTH is derived for two horizontal resolutions. If a cloud is detected at the native JSG resolution (1 pixel), it is considered to be optically thick and classified as *thick cloud* accordingly. If the cloud is found only after averaging signals horizontally (11-pixel gliding average), it is classified as *thin cloud*. If the uppermost cloud layer is penetrated by the laser beam and another cloud top is detected below, the classification is set to *multi-layer cloud* and, depending on the applied averaging, it is indicated whether a *thin over thick*, *thin over thin*, or *thick over thick* cloud layering was found.

The A-CTH product contains a quality indicator for the CTH determination in terms of a level of confidence. It is derived by comparing the obtained WCT at cloud top with the respective threshold value. Furthermore, since two independent methods for the detection of clouds are applied in the generation of ATLID profile and layer products, a level of consistency is provided. It is calculated by comparing the A-CTH values with the altitude of cloudy pixels in the A-TC product. The mathematical description of the algorithm is given in Sect. 3.3.

## 2.2 ALTID Aerosol Layer Descriptor (A-ALD)

The A-ALD product contains geometrical and optical information on aerosol layers. It provides the number of significant aerosol layers present in a profile, their individual upper and lower geometrical boundaries, layer-mean values of extinction and backscatter coefficient, lidar ratio, and particle linear depolarization ratio, the AOT of each layer, as well as the sum of the AOT of all layers, the stratospheric, and the columnar AOT. All optical parameters hold for the ATLID wavelength of 355 nm. A-ALD is defined for cloud-free conditions only, i.e., it is not generated when a cloud has been detected with the A-CTH algorithm. The product is provided with a horizontal resolution of 11 JSG pixels based on a gliding horizontal average along the satellite track. The five pixels next to clouds detected with the A-CTH algorithm are excluded to avoid cloud contamination of the aerosol product. The A-ALD product intrinsically contains the height of the planetary boundary layer (PBL), which is the top of the aerosol layer that is in contact with the ground. The Earth surface is the lower boundary of this aerosol layer by definition.

All optical data are supplemented with information on their statistical errors. As in the A-CTH product, a level of confidence for the layer detection and a level of consistency are provided. The latter one is derived by comparing the detected aerosol layers with the aerosol features from the A-TC product. For further use in the AM-ACD algorithm, vertically integrated columnar aerosol classification probabilities are stored in the product as well. They are calculated by weighting the aerosol classification probabilities from A-TC with the extinction coefficient from A-EBD. Details are explained in the mathematical description of the algorithm in Sect. 3.4.



# 3 ATLID Layer Products processor

## 3.1 Algorithm overview

The ATLID layer products A-CTH and A-ALD are generated sequentially by the ATLID Layer Products processor A-LAY. Fig. 1 shows the overall flowchart of the processor. The ATLID Mie co-polar signal (taken from A-NOM) is used to identify cloud and aerosol layer boundaries. With the help of information on the JSG provided in the X-JSG auxiliary product (Eisinger et al., 2023), the lidar profiles are resampled on the JSG and complemented with information on the height of the Earth surface taken from the A-TC product. Then, the algorithm searches for clouds, determines the CTH, assigns a cloud class, performs the consistency check against A-TC, and stores the results in the product file (left branch in Fig. 1). Aerosol layer information is calculated in the second step (right branch in Fig. 1), when no clouds were found over the number of JSG pixels selected for horizontal averaging (11 pixels by default). Next to input from A-EBD and A-TC, the tropopause height is needed. It is extracted from the auxiliary file with meteorological data (X-MET, Eisinger et al., 2023). The algorithm determines the aerosol layer boundaries, calculates the optical data and their errors, performs the consistency check against A-TC, computes the aerosol type probabilities, and then finishes with producing the output file. The sub-flows for A-CTH and A-ALD are further detailed in Sect. 3.3 and 3.4, respectively.

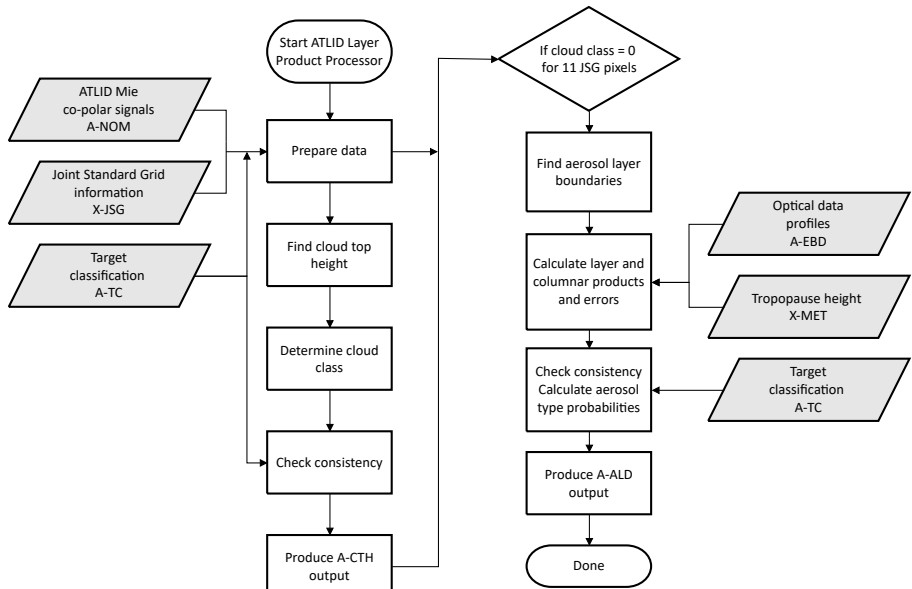

**Figure 1.** Flowchart of the ATLID Layer Products processor A-LAY.





## 3.2 Layer detection algorithm

### 3.2.1 Selection of the layer detection method

Cloud and aerosol layer detection from elastic-backscatter lidar signals has a long tradition and numerous methods have been proposed in the literature. Classical layer detection algorithms are based on the setting of thresholds, the search for vertical signal gradients, the detection of temporal/horizontal variances, as well as the application of WCT or image-processing techniques. Overviews and comparisons of the different methods, mainly with focus on determination of PBL heights from ground-based observations, have been presented, e.g., by Menut et al. (1999); Lammert and Bösenberg (2006); Baars et al. (2008); Emeis et al. (2008); Haeffelin et al. (2012); Toledo et al. (2017); Dang et al. (2019). Various studies have shown that for reliable operational algorithms, it is advisable to combine different methods and allow for adjustable parameter settings (e.g., Morille et al., 2007; Lewis et al., 2013; de Bruine et al., 2017; Bravo-Aranda et al., 2017; Kotthaus et al., 2020).

Compared to ground-based measurements, for which most of the algorithms have been developed, space-borne observations suffer from low SNR and coarse resolution. Therefore, traditional gradient and variance methods, which are sensitive to noise and require high spatial resolution, are not suited for space lidar applications. Threshold and WCT methods are much more robust and can be easily adjusted to actual observation conditions. Threshold methods have proven to be very useful for cloud detection and are widely used, e.g., for cloud-base determination with laser ceilometers from ground. Also the CALIPSO retrievals apply sophisticated threshold algorithms to detect cloud and aerosol layers (Vaughan et al., 2009; Winker et al., 2009). In these algorithms, threshold values vary as a function of target (cloud or aerosol), height, and horizontal resolution. In general, threshold values have to be carefully chosen in dependence on the achieved SNR to avoid misinterpretation of noise peaks (e.g., Chazette et al., 2001; Morille et al., 2007; Berthier et al., 2008).

The WCT technique allows for the analysis of signatures in signal profiles in a more sophisticated way (e.g., Cohn and Angevine, 2000; Davis et al., 2000; Brooks, 2003; Baars et al., 2008). Signal gradients are identified by measuring the similarity of the signal and a prescribed function, usually the Haar wavelet function (Haar, 1910). The parameters of the wavelet can be individually selected and adjusted for specific targets or situations. The technique is widely applied to determine the top of the PBL from automated ground-based lidar and ceilometer observations (e.g., Brooks, 2003; de Haij et al., 2006; Morille et al., 2007; Baars et al., 2008; Zhang et al., 2019). Wavelet analysis has also been used to detect the boundaries and internal structure of cirrus clouds (e.g., van den Heuvel et al., 2000; Wang and Sassen, 2006; Sassen et al., 2007; Wang and Sassen, 2008; Nakoudi et al., 2021) and polar stratospheric clouds (David et al., 2005). The WCT technique is often combined with certain threshold conditions. For instance, Morille et al. (2007) presented an automated algorithm to retrieve the vertical structure of the atmosphere, including clouds, aerosols, and molecular layers, by combining the WCT technique with threshold settings and a noise analysis of the signals.

Based on the literature survey and our experience with the automated processing of ground-based lidar network data (Baars et al., 2008; Pappalardo et al., 2014; Engelmann et al., 2016; Baars et al., 2016), a combined WCT and threshold technique has been chosen for the ATLID Layer Products processor. As shown below, the implementation is relatively simple and robust. A proper setting of WCT parameters and thresholds under consideration of the SNR of the ATLID Mie co-polar signal allows





for both cloud and aerosol layer detection with the same algorithm. The mathematical description of the WCT procedure is given in Sect. 3.2.2. The definition of thresholds is explained in Sect. 3.2.3. The specific algorithms and settings for CTH and
aerosol-layer detections are discussed in Sect. 3.3 and 3.4, respectively.

### 3.2.2  WCT algorithm

The WCT is defined as

$$W_\mathrm{f}(a,b) = \frac{1}{a} \int\limits_{z_\mathrm{b}}^{z_\mathrm{t}} f(z) h\left(\frac{z-b}{a}\right) \mathrm{d}z, \tag{1}$$

with the Haar function (Haar, 1910)

$$h\left(\frac{z-b}{a}\right) = \begin{cases} +1, \ b-\frac{a}{2} \leq z \leq b, \\ -1, \ b \leq z \leq b+\frac{a}{2}, \\ 0, \ \text{elsewhere.} \end{cases} \tag{2}$$

In Eq. (1), $f(z)$ is the signal to be analyzed (in our case the Mie co-polar signal), $z$ is the measurement height, and $z_\mathrm{b}$ and $z_\mathrm{t}$ are the bottom and top heights of the investigated profile, respectively. The Haar function is illustrated in Fig. 2a. It has two parameters. The dilation $a$ describes the extent of the Haar step function, or wavelet, which is 0.8 km in this case. The translation $b$ determines the actual location of the step, i.e., 11.3 km for the given example.

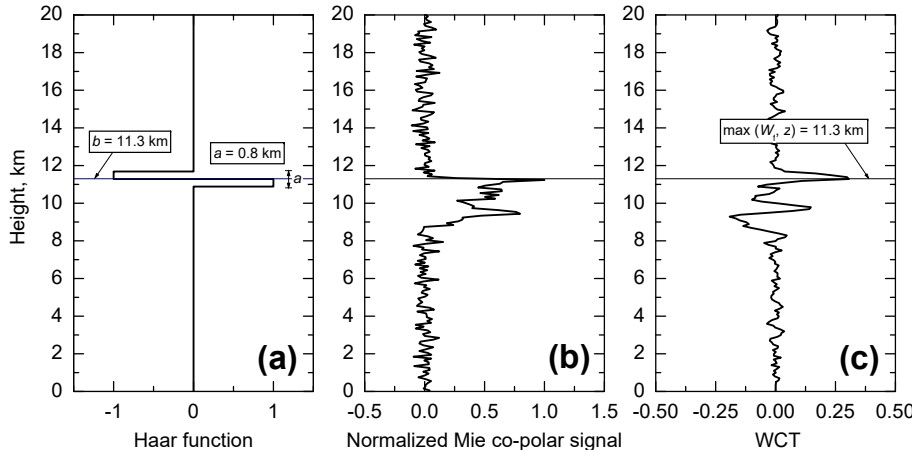

**Figure 2.** Principle of the WCT technique, a) Haar function with a = 0.8 km and b = 11.3 km, b) normalized Mie co-polar signal (simulated), c) WCT of the Haar function and the normalized Mie co-polar signal.

Figure 2b shows a simulated ATLID Mie co-polar signal, normalized to its maximum value, with a cirrus cloud feature between 8.8 and 11.3 km height (extinction profile taken from an observation with a ground-based lidar). The WCT is a measure of the similarity of the normalized lidar signal and the Haar function. For the calculation of the entire WCT profile





shown in Fig. 2c, the wavelet is slid along the signal profile by running the value of $b$ along $z$ from bottom to top. When the wavelet hits strong signatures (gradients) in the profile, $W_f(a, b)$ shows local extreme values. In the example of Fig. 2, the

absolute maximum of the WCT is found at the top of the cirrus cloud feature at 11.3 km.

The dilation $a$ of the Haar function is a configurable parameter. It determines how many data points are involved in the analysis, i.e., it has the role of a smoothing parameter. The optimal $a$ depends on the SNR of the lidar signal and thus varies depending on the target (clouds, aerosol layers) and measurement conditions (laser energy, atmospheric attenuation, background lighting, etc.).

For the A-LAY processor, a discrete formulation of the WCT algorithm is used. As mentioned above, the A-CTH and A-ALD products are provided on the JSG. Therefore, in the first step, the L1b Mie co-polar signals are resampled and averaged to the required horizontal resolution. The SNR of the averaged Mie co-polar signal $\bar{P}_{\text{Mie}}(z)$ with a horizontal resolution of either one or 11 JSG pixels and a vertical resolution of $\Delta z$ is calculated accordingly. The discrete WCT can then be written as:

$$W_f(n, b) = \frac{1}{n\Delta z}\left(\sum_{b-n\Delta z/2}^{b}\hat{P}_{\text{Mie}}(z)\Delta z - \sum_{b}^{b+n\Delta z/2}\hat{P}_{\text{Mie}}(z)\Delta z\right) = \frac{1}{n}\left(\sum_{b-n\Delta z/2}^{b}\hat{P}_{\text{Mie}}(z) - \sum_{b}^{b+n\Delta z/2}\hat{P}_{\text{Mie}}(z)\right). \quad (3)$$

Here, we have substituted the dilation $a$ in Eq. (1) and (2) by

$$a = n\Delta z, \text{ with } n = 2, 4, 6, 8, .... \quad (4)$$

To enable the application of thresholds to $W_f$, the normalized signal

$$\hat{P}_{\text{Mie}}(z) = \frac{\bar{P}_{\text{Mie}}(z)}{\max[\bar{P}_{\text{Mie}}(z)]} \quad (5)$$

is used in the calculation of the WCT. Then, the range of $W_f$ is [–0.5, 0.5]. The calculation is performed for heights $z$ between

the Earth surface and the uppermost JSG height, and the translation step width follows the vertical resolution of the signal $\Delta z$. The discrete values of $b$ have to be set in between two data points of $\hat{P}_{\text{Mie}}(z)$, so that each series in Eq. (3) contains the same number of terms. According to Eq. (3), $W_f$ is the difference of the mean values of the signal $\hat{P}_{\text{Mie}}(z)$ below and above the height of translation $b$ for layers of thickness $n\Delta z/2$. Fig. 3 illustrates the discrete WCT for idealized conditions.

### 3.2.3 Definition of thresholds

Thresholds need to be applied to decide whether a specific extreme value in the WCT profile can be assigned to a cloud or aerosol layer boundary (see, e.g., Fig. 2c). Two types of thresholds, $W_{\text{C/A},i}$ and $\text{SNR}_{\text{C/A},i}$, with $i$ different values are considered. $W_{\text{C/A},i}$, with $0 < |W_{\text{C/A},i}| < 0.5$, is the value that must be exceeded by the WCT. The threshold is positive for layer top heights and negative for base heights. $\text{SNR}_{\text{C/A},i}$ is the signal-to-noise ratio that is required for the Mie co-polar signal. It indicates whether a feature of a certain strength is present in the signal. If both thresholds are passed at the height where a local

extreme value in the WCT function is found, this height is considered as a potential layer boundary. The thresholds can be set independently for clouds (index C) and aerosol layers (index A). Furthermore, both thresholds can vary with height (index $i$). The algorithm allows individual settings for four height ranges: lower troposphere ($i = 0$), upper troposphere ($i = 1$),





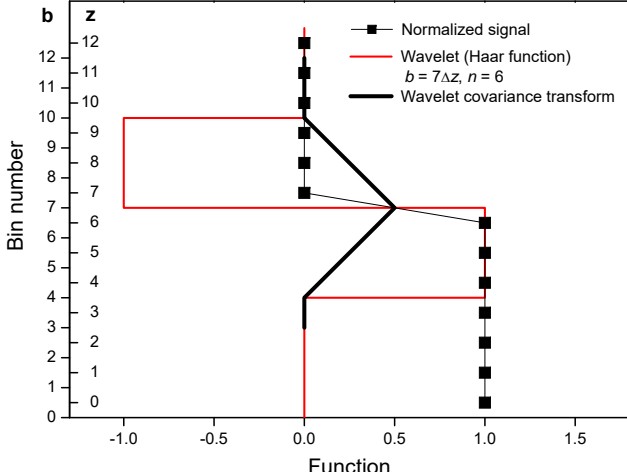

**Figure 3.** Principle of the WCT for discrete data points and idealized conditions. The symbols represent an idealized, normalized signal with values of 1 for height bins from 0 to 6 and values of 0 above. The wavelet (red line) with a dilation of $6\Delta z$ is translated to bin 7. At this position, the WCT (thick black line) takes the maximum possible value of 0.5.

stratosphere up to 20 km ($i = 2$), and stratosphere above 20 km ($i = 3$). The boundary between lower and upper troposphere depends on the height of the tropopause and is configurable. It can be set by dividing the height of the tropopause by the

troposphere partitioning parameter $p_{\text{trop}}$, which is defined as a floating-point number between 1.0 and 10.0 (default value 3.0). The boundary in the stratosphere is needed because of the change in vertical resolution, and thus SNR, of the Mie co-polar signal at 20 km height.

### 3.3 ATLID Cloud Top Height (A-CTH)

The algorithm for CTH detection is outlined in Fig. 4. After preparation of the input data (see Sect. 3.1), the scene is processed

twice, once with input signals averaged over one JSG pixel and once with input signals averaged over 11 JSG pixels. The WCT method with thresholds is applied to both data sets according to the parameter settings. The results are compared to make the decision on actual CTH and corresponding cloud class. Finally, the comparison with the A-TC cloudy pixels is performed to calculate the level of consistency and the results are stored in the A-CTH output. The steps are described in more detail in the following.

### 3.3.1 Detection of cloud top height

For the CTH retrieval, the WCT algorithm and the parameter settings described in Sect. 3.2 were further refined to account for realistic cloud scenes and noisy signals. First, an additional search loop was implemented to deal with multi-layer clouds. Here, it has to be considered that strong signals caused by lower clouds can dominate the wavelet analysis so that gradients in weaker signals from thin clouds above are not identified anymore. Therefore, after a CTH (i.e., the uppermost WCT maximum for





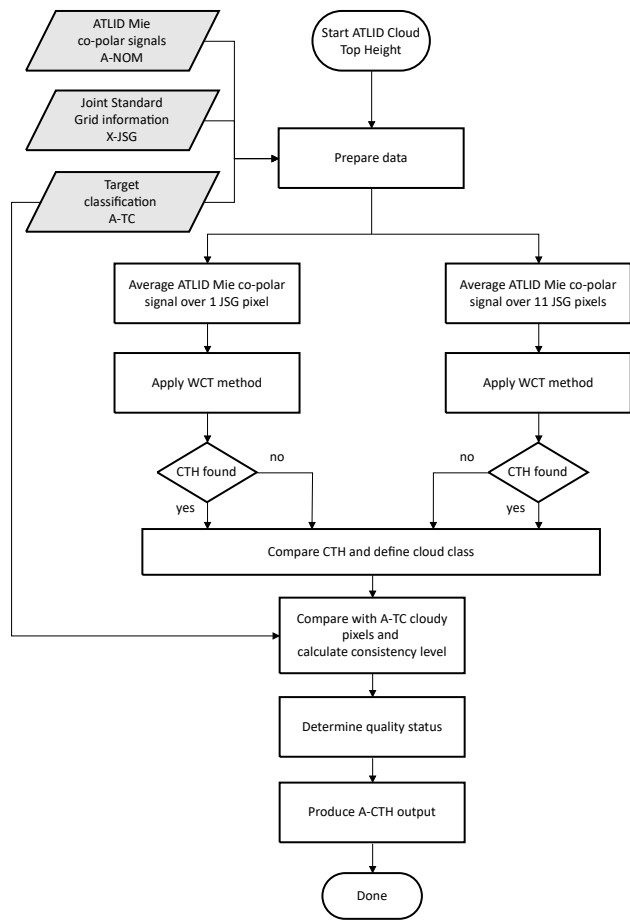

**Figure 4.** Flowchart for the A-CTH processing.

which the threshold conditions are met) was found from the search over the entire column, the altitude range from the surface up to this CTH is removed, a new normalization of the remaining signal is performed, and the WCT algorithm is applied again. If a new CTH is found, it replaces the former one. The search is repeated until no new CTH is found anymore.

As a second refinement, an additional configuration parameter $m_C$ was introduced to improve the performance of the algorithm in case of very noisy signals. It is used to smooth the SNR profile, i.e., the threshold $\text{SNR}_{C,i}$ is applied to the mean SNR over $m_C$ data points just below the potential CTH detected with the WCT (local maximum $> W_{C,i}$). As a typical value, $m_C = n/2$ can be chosen. Then, the wavelet and the noise analysis are performed with the same vertical resolution.

### 3.3.2 Quality indicator for CTH detection

A quality indicator in terms of a level of confidence is provided with the CTH. The level of confidence is determined from the actual value of $W_f(l_{\text{CTH}})$ at cloud top (range bin $l_{\text{CTH}}$) and the threshold value $W_{C,i}$ for this height. It is highest, if $W_f(l_{\text{CTH}})$ is





close to its maximum value of 0.5, and lowest, if $W_{\mathrm{f}}(l_{\mathrm{CTH}})$ is close to the threshold value. The value is defined as an integer on a linear scale from 1 to 10 as

$$C_{\mathrm{CTH}} = \mathrm{int}\left(10\frac{W_{\mathrm{f}}(l_{\mathrm{CTH}}) - W_{\mathrm{C},i}}{0.5 - W_{\mathrm{C},i}} + 0.99\right). \tag{6}$$

If no CTH was found, $C_{\mathrm{CTH}}$ is set to 0.

### 3.3.3 Simplified classification of uppermost cloud

The algorithm as described above detects optically thick clouds on the basis of Mie co-polar signals averaged over one JSG pixel. For the detection of optically thin clouds, a gliding horizontal average of 11 JSG pixels is applied. The algorithm is run successively for both resolutions. Then, the results are compared for each JSG pixel. To indicate the presence of different cloud types for further use in the synergistic CTH algorithm (AM-CTH product, Haarig et al., 2023), a simplified cloud class $F_{\mathrm{CTH}}$ is assigned to the pixel as follows:

- **$F_{\mathrm{CTH}}$ = 0, no cloud:** No CTH is found, neither with one nor with 11 JSG pixel horizontal resolution. The pixel is indicated as cloud-free.

- **$F_{\mathrm{CTH}}$ = 1, thick cloud**: The CTH is found with one JSG pixel horizontal resolution and no higher boundary is found with 11 JSG pixel resolution. An optically thick cloud is present.

- **$F_{\mathrm{CTH}}$ = 2, thin cloud**: The CTH is found with 11 JSG pixel horizontal resolution, but not with one JSG pixel horizontal resolution. An optically thin cloud is present.

- **$F_{\mathrm{CTH}}$ = 3, multiple cloud layers, thin over thick**: The CTH found with 11 JSG pixel horizontal resolution is higher than the CTH found with one JSG pixel horizontal resolution. An optically thin cloud above an optically thick cloud is present.

- **$F_{\mathrm{CTH}}$ = 4, multiple cloud layers, thick over thick**: At least two different cloud upper boundaries are found with one JSG pixel horizontal resolution and no higher boundary is found with 11 JSG pixel resolution. At least two optically thick cloud layers are present.

- **$F_{\mathrm{CTH}}$ = 5, multiple cloud layers, thin over thin**: At least two different cloud upper boundaries are found with 11 JSG pixel horizontal resolution, but no CTH is found with one JSG pixel resolution. At least two optically thin cloud layers are present.

- **$F_{\mathrm{CTH}}$ = 6, no cloud, but probably cloud influenced**: No CTH is found, neither with one nor with 11 JSG pixel horizontal resolution, but the profile contributed to the closest detected thin cloud. This class is used to exclude the five pixels next to a thin cloud (found with 11 JSG pixel horizontal resolution) from the aerosol layer detection (see Sect. 3.4) as these pixels are probably influenced by the cloud boundaries.





To assign the multi-layer cloud classes $F_{\mathrm{CTH}} = 3,\ 4,\ 5$, an optional test is implemented in addition, which checks whether the different CTH belong to layers that are separated by a clear-air gap. For this purpose, the SNR profile is smoothed over $m_{\mathrm{C}}$ height bins and the threshold $\mathrm{SNR}_{\mathrm{C},i}$ is applied to each bin between the different CTH to decide whether it is cloudy or cloud-free. If more than a configurable number of bins are found to be cloud-free, two distinct cloud layers are detected. If no gap is found, the layer boundaries are considered as internal cloud structure and $F_{\mathrm{CTH}} = 3,\ 4$ is changed to $F_{\mathrm{CTH}} = 1$ (thick cloud) and $F_{\mathrm{CTH}} = 5$ to $F_{\mathrm{CTH}} = 2$ (thin cloud).

### 3.3.4 Consistency of A-CTH with A-TC

The derived CTH is compared with the uppermost cloudy pixel identified by the A-PRO algorithm (stored in the A-TC product). A two-dimensional level of consistency $X_{\mathrm{CTH}} = (i,\ j)$, with $i = 0 \ldots 3$ and $j = 0 \ldots 10$, is defined as follows:

- **$X_{\mathrm{CTH}}$ = (0, 0):** No cloud is present neither in A-CTH nor in A-TC.

- **$X_{\mathrm{CTH}}$ = (1, 0):** Cloud is present in A-CTH, but not in A-TC.

- **$X_{\mathrm{CTH}}$ = (2, 0):** Cloud is present in A-TC, but not in A-CTH.

- **$X_{\mathrm{CTH}}$ = (3, $j$):** Cloud is present in A-CTH and A-TC and
  $j = 1 \ldots 10$, if $\Delta z_{\mathrm{CTH}} < c_{\mathrm{C}}(11 - j)$, else $j = 0$,
  with the configurable confidence criterion $c_{\mathrm{C}}$ and the CTH difference $\Delta z_{\mathrm{CTH}}$ between the A-CTH and A-TC products.

For instance, if $c_{\mathrm{C}}$ is set to $100\,\mathrm{m}$, then $j = 10$ means that the CTH found by both algorithms is within one height bin (difference $< 100\,\mathrm{m}$). Each decrease of the consistency level by $\Delta j = 1$ indicates an increase of the CTH difference by about $100\,\mathrm{m}$, and a level of consistency of $X_{\mathrm{CTH}} = (3,\ 0)$ shows that the CTH difference is $\geq 1000\,\mathrm{m}$.

### 3.3.5 A-CTH quality status

The A-CTH quality status summarizes the results of the confidence and consistency checks described in Sect. 3.3.2 and 3.3.4, respectively, in a single value from 0 (highest quality) to 4 (bad quality). A quality status of $-1$ is used when no cloud was detected in the profile. The quality status is defined as follows:

- **$Q_{\mathrm{CTH}}$ = 0:** The data are of good quality.

- **$Q_{\mathrm{CTH}}$ = 1:** The data are valid, but the level of confidence is lower than the configurable threshold $q_{\mathrm{C,loc}}$ (default value 5).

- **$Q_{\mathrm{CTH}}$ = 2:** The data are valid, but the derived CTH is not consistent with the A-TC product. The consistency check led to $j < q_{\mathrm{C,con}}$, where $q_{\mathrm{C,con}}$ is a configurable threshold (default value 3).

- **$Q_{\mathrm{CTH}}$ = 3:** The data are valid, but there is no cloud in the A-TC product.

- **$Q_{\mathrm{CTH}}$ = 4:** The (input) data are invalid.

- **$Q_{\mathrm{CTH}}$ = –1:** No cloud was detected.





## 3.4 ATLID Aerosol Layer Descriptor (A-ALD)

Figure 5 shows the flowchart of the A-ALD processing. For the preparation of the input data (see Sect. 3.1), the information on
300  cloud-free pixels from the A-CTH part of the A-LAY processor is needed. The same horizontal averaging as for the detection
of thin clouds is used, i.e., the algorithm is applied to groups of 11 JSG pixels, for which $F_{\mathrm{CTH}} = 0$ was determined with the
A-CTH algorithm (see Sect. 3.3.3). A gliding search with one pixel resolution is performed. The WCT method with thresholds
is adopted to search for aerosol layer boundaries. Then, the mean optical data for each detected layer and the AOT values are
calculated. Similar as for the A-CTH product, a comparison with A-TC is performed and a level of consistency is provided. In
305  addition, the columnar aerosol classification probabilities are computed. All variables are stored with the respective error and
quality information in the A-ALD output. Details are described in the following.

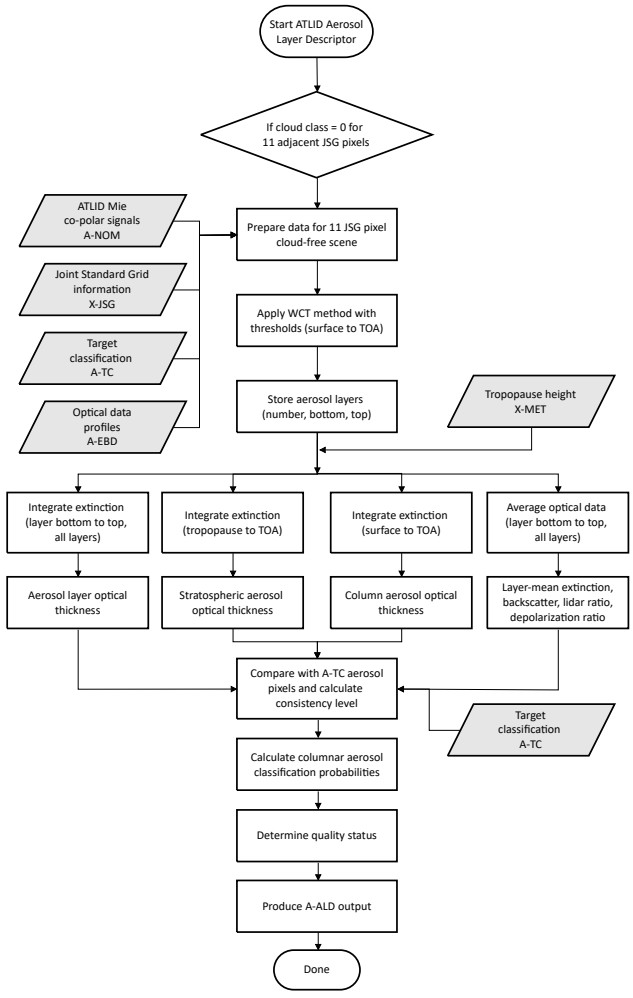

**Figure 5.** Flowchart for the A-ALD processing.





### 3.4.1 Detection of aerosol layers

For the search of aerosol layer boundaries, the WCT algorithm with thresholds is applied to the horizontally averaged, cloud-free Mie co-polar signals under consideration of the specific parameter settings for aerosol. Since both base and top heights are needed, the WCT threshold actually consists of a pair of positive and negative values. Thus, after calculation of the WCT profile, the algorithm searches for all local maxima/minima of the WCT that are larger/smaller than the predefined threshold values $+W_{A,i}/-W_{A,i}$, respectively. The maxima are stored as potential layer top heights and the minima as potential layer base heights. The length of the window that is used for the search (in which not more than one maximum and one minimum can be found) is equal to the dilation of the WCT. The Earth surface is added as a potential layer base, if not automatically detected by the WCT search. Thus, the existence of the planetary boundary layer (PBL) as the aerosol layer that is in touch with the surface is presumed. However, the PBL is not necessarily detected by the algorithm, which may be the case, e.g., when the aerosol load is too low or the signal is strongly attenuated in lofted layers.

After the potential layer boundaries are identified, the algorithm checks the presence of aerosol in the potential layers. For this purpose, the mean SNR of the Mie co-polar signal between any two successive potential layer boundaries is calculated and compared to the predefined threshold value $SNR_{A,i}$. If the threshold is exceeded, the layer is stored with its respective top and bottom heights.

After the entire scene is processed, an additional filter is applied to remove falsely detected layers. Such false detection can happen, if strong noise peaks occur that are misinterpreted as layers because the thresholds are passed. Noise peaks are singular events, which influence only one or few adjacent profiles (due to horizontal averaging). Thus, by assuming that real aerosol layers show coherent structures on a larger scale, the falsely detected layers can be removed. The filter checks for each detected layer the five neighboring profiles to each side. If at least four layer bases or layer tops are detected within $\pm 1$ vertical bin distance from the aerosol layer base or layer top, respectively, the aerosol layer is considered to be true and not induced by noise. In case the filter criteria are not fulfilled, the aerosol layer is removed.

A further feature of the algorithm is that it can be configured to either detect or not detect the internal structure of aerosol layers (or layers attached to each other). If internal layers are allowed, any layer top, except the uppermost, can be a layer base as well and any layer base, except the surface, can be a layer top as well. When the detection of internal aerosol layers is suppressed, each layer boundary is either a layer top or a layer base, i.e., the detected layers are always separated by a clear-air gap.

### 3.4.2 Quality indicators for aerosol layer detection

Quality indicators in terms of a level of confidence are defined for the detection of aerosol layers and layer boundaries. The level of confidence for the identification of layer boundaries is determined from the actual value $W_f(l_{B,T})$ of the WCT at the boundary, i.e., the local minimum or maximum at range bin $l_{B,T}$ of a layer base (index B) or top (index T), respectively. It is highest, if $|W_f(l_{B,T})|$ is close to its maximum value of 0.5, and lowest, if $|W_f(l_{B,T})|$ is close to the threshold value $|W_{A,i}|$. The





value is defined as an integer on a linear scale from 1 to 10 as

$$C_{\mathrm{B,T}} = \mathrm{int}\left(10\frac{|W_{\mathrm{f}}(l_{\mathrm{B,T}})| - |W_{\mathrm{A},i}|}{0.5 - |W_{\mathrm{A},i}|} + 0.99\right). \tag{7}$$

In a similar way, the mean signal-to-noise ratio $\overline{\mathrm{SNR}}$ of the Mie co-polar signal within a layer (between range bins $l_{\mathrm{B}}$ and $l_{\mathrm{T}}$) is used to define the level of confidence for the presence of a distinct aerosol layer:

$$C_{\mathrm{L}} = \mathrm{int}\left(9\frac{\overline{\mathrm{SNR}} - \mathrm{SNR}_{\mathrm{A},i}}{10 - \mathrm{SNR}_{\mathrm{A},i}} + 0.99\right), \ \text{if } \overline{\mathrm{SNR}} \leq 10, \ \text{and } C_{\mathrm{L}} = 10, \ \text{if } \overline{\mathrm{SNR}} > 10. \tag{8}$$

$\mathrm{SNR}_{\mathrm{A},i}$ is the SNR threshold value used for the layer detection. The highest level of confidence is reached, if the mean SNR is
above 10.

### 3.4.3 Columnar and layer optical properties

Optical properties of the detected aerosol layers and the entire aerosol column are derived from the profiles of optical variables, which are provided in the A-EBD product with three different resolutions (Donovan et al., 2023a). Preferably, the high-resolution data (per JSG pixel) are used and averaged over the 11 JSG pixels, for which the layer boundaries were de-
termined. Alternatively, the user can select the medium- or low-resolution A-EBD data, which will then be used as is without further averaging. In the latter case, the retrieved optical data may have a better quality (less noise), but they may not be fully consistent with the layer boundaries because of the different resolution.

The column AOT at the ATLID wavelength of 355 nm is calculated by integrating the profile of the particle extinction coefficient $\alpha_{355}(z)$ from the surface ($z_{\mathrm{S}}$) to the top of the atmosphere ($z_{\mathrm{TOA}}$):

$$\mathrm{AOT}_{355} = \int_{z_{\mathrm{S}}}^{z_{\mathrm{TOA}}} \alpha_{355}(z)\mathrm{d}z = \sum_{k=l_{\mathrm{S}}+1}^{l_{\mathrm{top}}} \alpha_{355}(z_k)\Delta z_k. \tag{9}$$

The integration is replaced by the summation of discrete extinction values in the height bins of width $\Delta z_k$. The summation starts with the first height $z_{\mathrm{S}+1}$ (at height bin $l_{\mathrm{S}} + 1$) that is not influenced by the surface return. The top of the atmosphere is replaced by the uppermost retrieval height $z_{\mathrm{top}}$ (at height bin $l_{\mathrm{top}}$) of $\alpha_{355}(z_k)$. Accordingly, the stratospheric AOT is computed by integrating the extinction profile from the tropopause (obtained from X-MET) to the uppermost retrieval height. Further-
more, the aerosol layer optical thickness is calculated for each detected layer using the layer boundaries determined before (Sect. 3.4.1). In addition to the column aerosol optical thickness, also the sum of aerosol layer optical thicknesses is provided. The difference between the two values indicates how much aerosol is distributed in between the detected distinct aerosol layers.

The A-ALD product also contains, for each detected layer, the layer-mean values of particle extinction coefficient, backscatter coefficient, lidar ratio, and linear depolarization ratio and their errors. Errors for all columnar and layer-mean optical data are
calculated from the individual errors provided in the A-EBD product. Error propagation is considered for horizontal averaging, when the high-resolution data are used as input, and for vertical averaging (or integration) for all kinds of input data.



### 3.4.4 Consistency of A-ALD with A-TC

The derived aerosol layers are compared with the aerosol pixels identified by the A-PRO algorithm (stored in the A-TC product). A two-dimensional level of consistency $X_{\text{AER}} = (i,\ j)$, with $i = 0\ldots 3$ and $j = 0\ldots 10$, is defined as follows:

- **$X_{\text{ALD}}$ = (0, 0):** The profile is not cloud-free, neither in A-ALD nor in A-TC.

- **$X_{\text{ALD}}$ = (1, 0):** The profile is cloud-free in A-ALD, but not in A-TC.

- **$X_{\text{ALD}}$ = (2, 0):** The profile is cloud-free in A-TC, but not in A-ALD.

- **$X_{\text{ALD}}$ = (3,$j$):** The profile is cloud-free in A-ALD and A-TC and

  $j = 1,\ 2,\ldots,\ 9,\ 10$, if $r_{\text{a}} > 0.1, 0.2, \ldots, 0.9, 0.99$, else $j = 0$,

with the agreement ratio $r_{\text{a}}$, which is the number of height bins of equal detection status (aerosol or clean) in the A-ALD and A-TC products divided by the total number of height bins.

For instance, if $j = 10$, then both algorithms identified $> 99\,\%$ of the height bins equally as either clean or loaded with aerosol. If $j = 9$, then both algorithms show more than $90\,\%$ agreement, and then each decrease of the consistency level by $\Delta j = 1$ indicates a decrease in agreement by $10\,\%$.

### 3.4.5 Columnar aerosol classification probability

Height-resolved aerosol classification probabilities are calculated with the A-PRO processor and stored in the A-TC product for mixtures of seven aerosol types (dust, marine aerosol, continental pollution, smoke, dusty smoke, dusty aerosol mix, ice; Donovan et al., 2023a). The aerosol classification follows from the Hybrid End-to-End Classification (HETEAC) model developed for EarthCARE (Wandinger et al., 2022). The ice is considered to indicate the presence of optically thin ice-containing 385  layers (e.g., diamond dust, subvisible cirrus) that have not been identified as clouds and thus occur in the aerosol products. The seven aerosol type probabilities $p_m$ ($m = 1...7$) are weighted with the extinction coefficient $\alpha_{355,\text{med}}$ at medium resolution for each height interval and integrated over the entire column to estimate the absolute contribution $\text{AOT}_{355,m}$ of each type $m$ to the total AOT:

$$\text{AOT}_{355,m} = \sum_{k=l_{\text{S}}+1}^{l_{\text{top}}} p_m(z_k)\alpha_{355,\text{med}}(z_k)\Delta z_k. \tag{10}$$

The columnar aerosol classification probability $P_{355,m}$ describes the relative contribution of each of the seven types to the AOT measured with ATLID:

$$P_{355,m} = \frac{\text{AOT}_{355,m}}{\sum_{m=1}^{7} \text{AOT}_{355,m}}. \tag{11}$$

The columnar aerosol classification probability is an important input parameter for the AM-COL algorithm (Haarig et al., 2023), which combines the ATLID aerosol typing with the MSI aerosol classification provided in the M-AOT product (Docter 395  et al., 2023).



### 3.4.6 A-ALD quality status

The A-ALD quality status summarizes the results of the confidence and consistency checks described in Sect. 3.4.2 and 3.4.4, respectively. It is provided for each JSG pixel (not for each aerosol layer) on a scale from 0 (highest quality) to 4 (bad quality). A quality status of $-1$ is used when a cloud was detected in the profile. The quality status is defined as follows:

- $Q_{\text{ALD}}$ **= 0:** The data are of good quality.

- $Q_{\text{ALD}}$ **= 1:** The data are valid, but the number of aerosol layers is higher than the configurable threshold $q_{\text{A,nal}}$ (default value 2).

- $Q_{\text{ALD}}$ **= 2:** The data are valid, but the relative uncertainty of the mean backscatter coefficient of at least one aerosol layer is larger than the configurable threshold $q_{\text{A,bsc}}$ (default value 0.1). Furthermore, the consistency level $j$ is smaller than
the configurable threshold $q_{\text{A,con}}$ (default value 8).

- $Q_{\text{ALD}}$ **= 3:** Warning that data are provided, but the A-TC product indicates a cloud in the profile.

- $Q_{\text{ALD}}$ **= 4:** The (input) data are invalid.

- $Q_{\text{ALD}}$ **= –1:** A cloud was detected in the profile.

## 4 Algorithm tests with simulated data

Atmospheric test scenes created from numerical model output data have been used for developing, testing, and evaluating the entire chain of EarthCARE processors. Cloud and precipitation information is based on output of the Global Environmental Multiscale (GEM) model, while aerosol fields were taken from the Copernicus Atmosphere Monitoring Service (CAMS) model (Qu et al., 2022b). By applying a sophisticated instrument simulator (Donovan et al., 2023b), test data for three dedicated scenes representing typical EarthCARE processing frames of one eighth of an orbit were generated. According to geographic locations
that are crossed by the simulated satellite tracks, the test scenes are called *Halifax*, *Baja*, and *Hawaii*. A detailed description of the scene selection, generation, and contents is provided by Qu et al. (2022b). In the following, selected results obtained with the A-LAY processor for the test scenes are presented to demonstrate the performance of the A-CTH (Sect. 4.1) and A-ALD (Sect. 4.2) algorithms.

### 4.1 A-CTH algorithm tests

Figure 6 shows the 2D cross section of the simulated ATLID Mie co-polar signal for the *Halifax* scene. The cloud top heights determined with the A-CTH algorithm are overlaid as black squares. The color code for the logarithmic representation of the L1b data is adopted from the well-known CALIPSO lidar browse images (see, e.g., respective products on the CALIPSO website, https://www-calipso.larc.nasa.gov/products/). Strongly scattering targets such as dense clouds appear in white and gray. Red and yellow colors indicate weaker backscatter signals caused by aerosols and optically thin clouds or by attenuation of



the signal in optically thick clouds. The bluish background represents either clear air or regions where the signal is completely attenuated by thick clouds above. The *Halifax* scene stretches from Greenland over the eastern end of Canada towards the Caribbean. The northern part of the scene (68° to 48° N) is dominated by snowing ice and mixed-phase clouds. A convective system with multiple high-reaching cloud layers related to a precipitating cold front starts over eastern Canada (at about 46° N) and reaches southward to Bermuda (at about 33° N). In the southern part of the scene, scattered low-level cumulus clouds are
embedded in a weak aerosol layer.

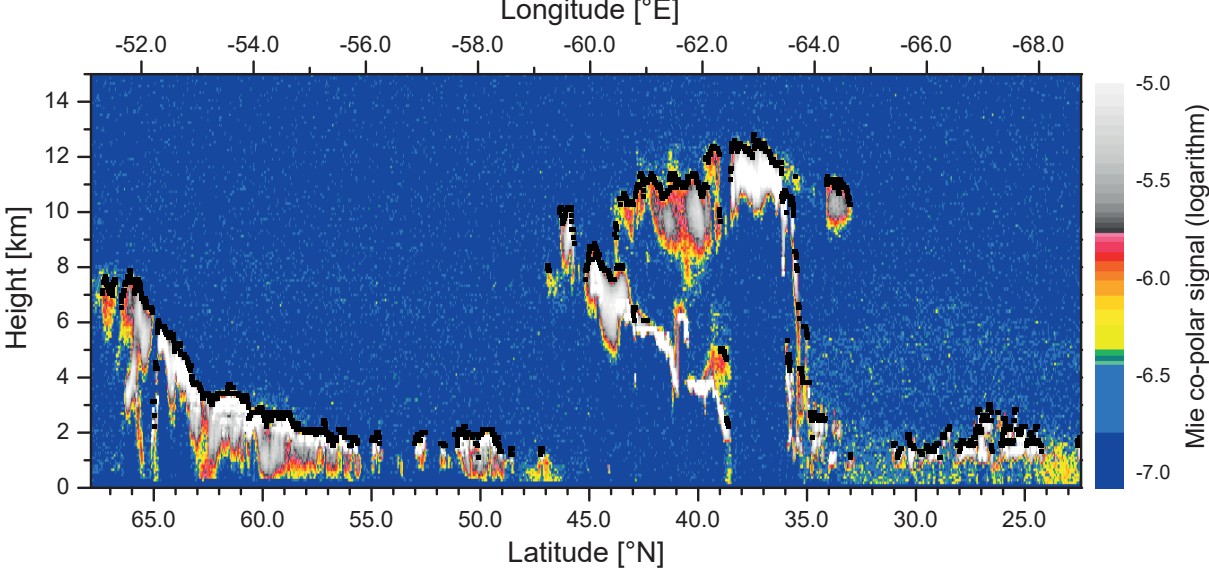

**Figure 6.** Cross section of the simulated ATLID Mie co-polar signal overlaid with cloud top heights from the A-CTH product (black squares) for the *Halifax* scene. The surface return has been removed and horizontal smoothing over 11 JSG pixels has been applied for plotting the L1b signals.

      In Fig. 7, the A-CTH product is presented in more detail for the middle part of the *Halifax* scene (convective system). Table 1 lists the configuration parameters applied in the processing. As can be seen from the first panel below the 2D cross section, all defined cloud classes ($F_{CTH}$, see Sect. 3.3.3) appear in this part of the scene. Each vertical bar on the colored horizontal lines, which stand for the cloud classes (see description to the right), represents a single JSG pixel. The high-reaching precipitating
ice clouds of the convective system (around 36°–38.7°, 44°–45°, and 46°–46.5° N) are identified as thick clouds, i.e., on the native JSG grid without averaging. In such optically dense ice clouds, the signals are often completely attenuated after 2–3 km penetration depth, as in the case between 36° and 38.5° N where no structures below 10 km can be seen. The top of the cirrus at about 40°–40.5° N is also detected at the highest resolution and thus indicated as belonging to a thick cloud. However, this cirrus layer is semi-transparent for the laser beam and the optically thick water cloud with a top at 4 km height, which finally
attenuates the signal completely, is identified as well. Consequently, the classification is "thick over thick" for this part of the



scene. Most of the other ice clouds require averaging over 11 JSG pixels to determine the CTH and are thus either classified as single-layer "thin" (see, e.g., 33°–34° N) or multi-layer "thin over thick" clouds (dominating in the range from 40.5°–44° N).

The lower three panels of Fig. 7 provide some information on the quality of the CTH determination. The level of confidence ($C_{\mathrm{CTH}}$, see Sect. 3.3.2) varies between 1 and 9 and often shows low values of 1 or 2, which means that the obtained WCT
maximum just passed the chosen threshold value and thus the signal gradient at the cloud top is relatively weak (see Sect. 3.3.2). As can be seen from the 2D cross section, regions of increased scattering are often present above the derived CTH for the high clouds. These features make it difficult to determine a clear cloud boundary. In these regions, also the consistency with the A-TC product ($X_{\mathrm{CTH}}$, see Sect. 3.3.4) is lowest. The A-TC product is based on a layer approach with certain horizontal averaging (Irbah et al., 2022; Donovan et al., 2023a). It typically has a lower resolution than A-CTH, i.e., the target boundaries, and
thus also the cloud top heights, appear smoother than in the A-CTH product. Sometimes, cloud gaps found in A-CTH are not present in A-TC (see the level of consistency with $X_{\mathrm{CTH}} = (2, 0)$, dark red bars and blue dots, respectively). The quality status ($Q_{\mathrm{CTH}}$, see Sect. 3.3.5) shown in the lowermost panel reflects these findings according to the threshold settings (see Table 1).

**Table 1.** Configuration parameters used in the processing of the test scenes with the A-CTH algorithm.

| Configuration parameter | Symbol | Value |
|---|---|---|
| Dilation of the wavelet | $n_{\mathrm{C}}$ | 6 |
| Troposphere partitioning parameter | $p_{\mathrm{trop}}$ | 3 |
| WCT threshold, lower troposphere | $W_{\mathrm{C},1}$ | 0.05 |
| WCT threshold, upper troposphere | $W_{\mathrm{C},2}$ | 0.05 |
| WCT threshold, stratosphere below 20 km | $W_{\mathrm{C},3}$ | 0.05 |
| WCT threshold, stratosphere above 20 km | $W_{\mathrm{C},4}$ | 0.05 |
| Number of bins for SNR smoothing | $m_{\mathrm{C}}$ | 3 |
| SNR threshold, lower troposphere | $\mathrm{SNR}_{\mathrm{C},1}$ | 15 |
| SNR threshold, upper troposphere | $\mathrm{SNR}_{\mathrm{C},2}$ | 5 |
| SNR threshold, stratosphere below 20 km | $\mathrm{SNR}_{\mathrm{C},3}$ | 5 |
| SNR threshold, stratosphere above 20 km | $\mathrm{SNR}_{\mathrm{C},4}$ | 5 |
| Minimum bin number for clear-air gap | $\Delta z_{\mathrm{ml}}$ | 5 |
| Consistency criterion [m] | $c_{\mathrm{C}}$ | 100 |
| Quality threshold for level of confidence | $q_{\mathrm{C,loc}}$ | 3 |
| Quality threshold for consistency with A-TC | $q_{\mathrm{C,con}}$ | 5 |

Figure 8 shows a direct comparison of the derived CTH values with the model truth. Next to the *Halifax* scene (Fig. 8a), results are also presented for the *Baja* and *Hawaii* scenes. The *Baja* scene (Fig. 8b) starts over northern Canada, crosses the
Rocky Mountains, and ends over the Baja California peninsula. The scene comprises very clear conditions in the northern part, scattered clouds over the Canadian Prairies, overcast over the Rocky Mountains, cloud-free conditions with high aerosol load over Utah and Arizona, and cirrus clouds in the southern part. The *Hawaii* scene (Fig. 8c) extends over the tropical Pacific



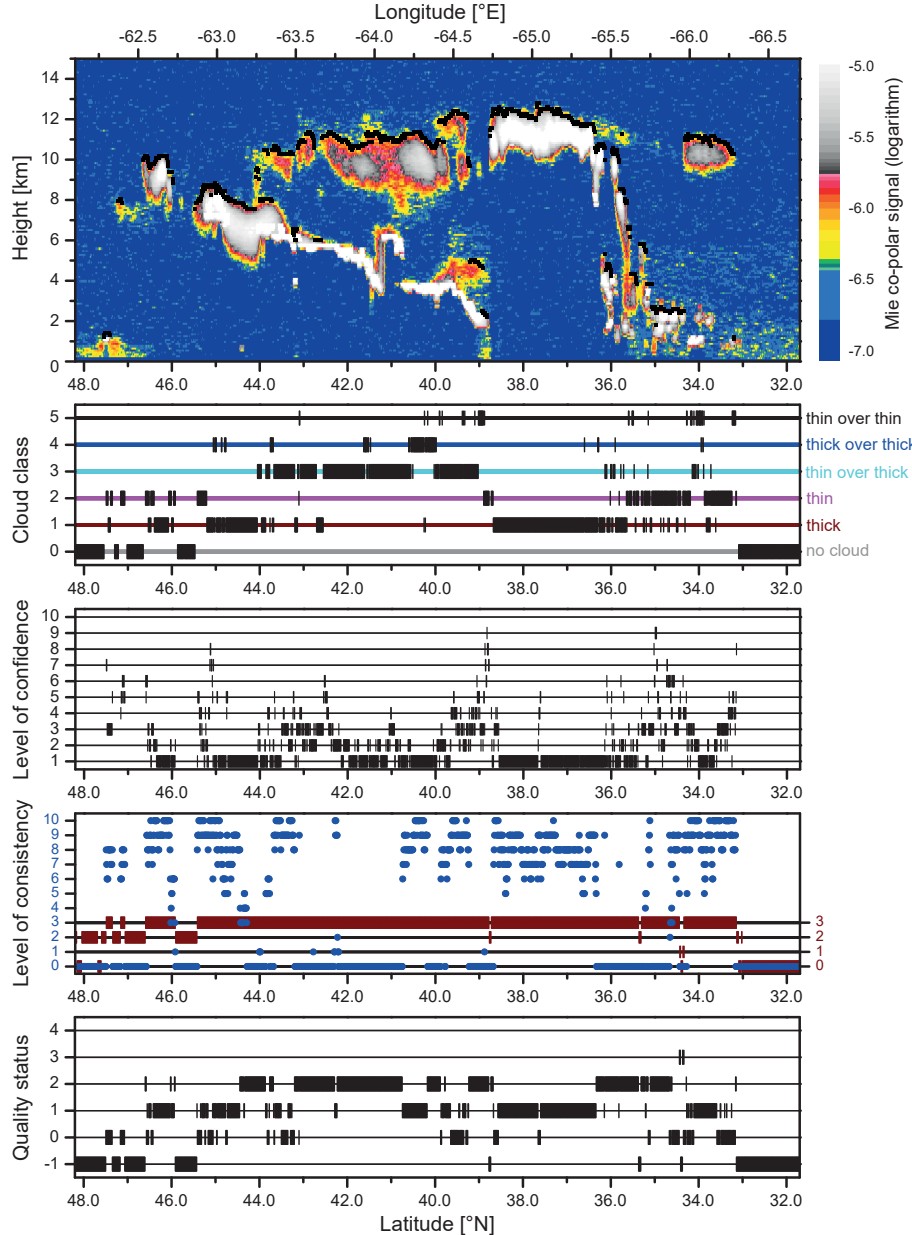

**Figure 7.** Illustration of the A-CTH product for the middle part of the *Halifax* scene. The same settings as for Fig. 6 are used for the 2D cross section in the uppermost panel. The panels below show the cloud classification ($F_{CTH}$), the CTH level of confidence ($C_{CTH}$), the level of consistency with the A-TC product ($X_{CTH} = (i, j)$, with $i$ in dark red and $j$ in blue), and the CTH quality status ($Q_{CTH}$). The processing was performed with the parameter settings given in Table 1.





across the Hawaiian Islands. The scene is dominated by a tropical convective system and high ice clouds reaching up to almost 18 km height. For the comparison, the simulated extinction fields for cloud hydrometeors (see Qu et al., 2022b) were used and

a threshold value was applied to determine the "true" CTH. The extinction threshold was varied between 1 and 50 $Mm^{-1}$. Best agreement in terms of minimum difference between A-CTH and model truth was found for a threshold value of 20 $Mm^{-1}$, for which the results are presented in Fig. 8. In this case, about two-thirds of the CTH values agree within ±300 m and 87 % within ±600 m for all scenes (see the numbers provided in the scatter plots). As expected, the largest differences are found for optically thin cirrus clouds in the 8–14 km height range of the *Halifax* and *Hawaii* scenes. Moreover, it was found that with

the chosen threshold settings (see Table 1), the algorithm often fails to detect thin or scattered clouds in the lower troposphere. Pixels with undetected clouds are indicated with small vertical bars on top of the x-axis in the left panels of Fig. 8. On average, they make up about 11 % of all cloudy model pixels. For about 3 % of all test-scene pixels, clouds are reported in the A-CTH product although there is no cloud in the model truth. These cases are mainly related to the 11-pixel averaging across cloud edges, which can artificially stretch the appearance of clouds horizontally.

In general, when judging the quality of CTH detection as done in Fig. 7 and 8, one has to keep in mind that the cloud structures in the test scenes originate from a numerical forecast model, and thus the shape of the cloud boundaries may not always be fully realistic. Furthermore, a much wider range of cloud–aerosol scenarios is needed for optimizing the threshold settings to properly discriminate clouds and aerosol in different heights and geographical locations. Therefore, it will be important to carefully test and validate the CTH algorithm with real-world data after the EarthCARE launch and to adapt the configuration

parameters accordingly.

## 4.2   A-ALD algorithm tests

Because the three standard test scenes are strongly dominated by clouds, the modified *Halifax aerosol* scene has been generated for testing the aerosol-related algorithms. The *Halifax aerosol* scene is a shorter test scene representing the southern 2000 km of the *Halifax* scene. The extinction of the marine aerosol type has been increased by a factor of 2.5, whereas all other aerosol

types and the liquid clouds have been downscaled by a factor of $10^{-6}$. The resulting 2D cross section of the simulated ATLID Mie co-polar signal is shown in the upper panels of Fig. 9 and 10. Ice clouds are still present north of 33° N, but the major part of the scene is dominated by a distinct marine boundary layer. The base and top heights of this layer as derived with the A-ALD algorithm are overlaid in gray and black, respectively, on the color plots. In the panels below the 2D cross sections, different variables contained in the A-ALD product are plotted. The configuration parameters applied in the processing are

listed in Table 2.

The second panel of Fig. 9 shows the comparison of the derived columnar AOT (blue) with the model truth (orange). In general, the agreement is very good, although an increase in the noise levels of the retrieved AOT values in the southern part of the scene is evident. The noise is linked to the decreasing SNR in the Rayleigh signal, which is due in part to the increasing solar background in this section of the scene. In addition, the increasing particle backscattering in the marine boundary layer

decreases the SNR of the Rayleigh signal via the correction of the cross-talk between the Rayleigh and Mie channels.





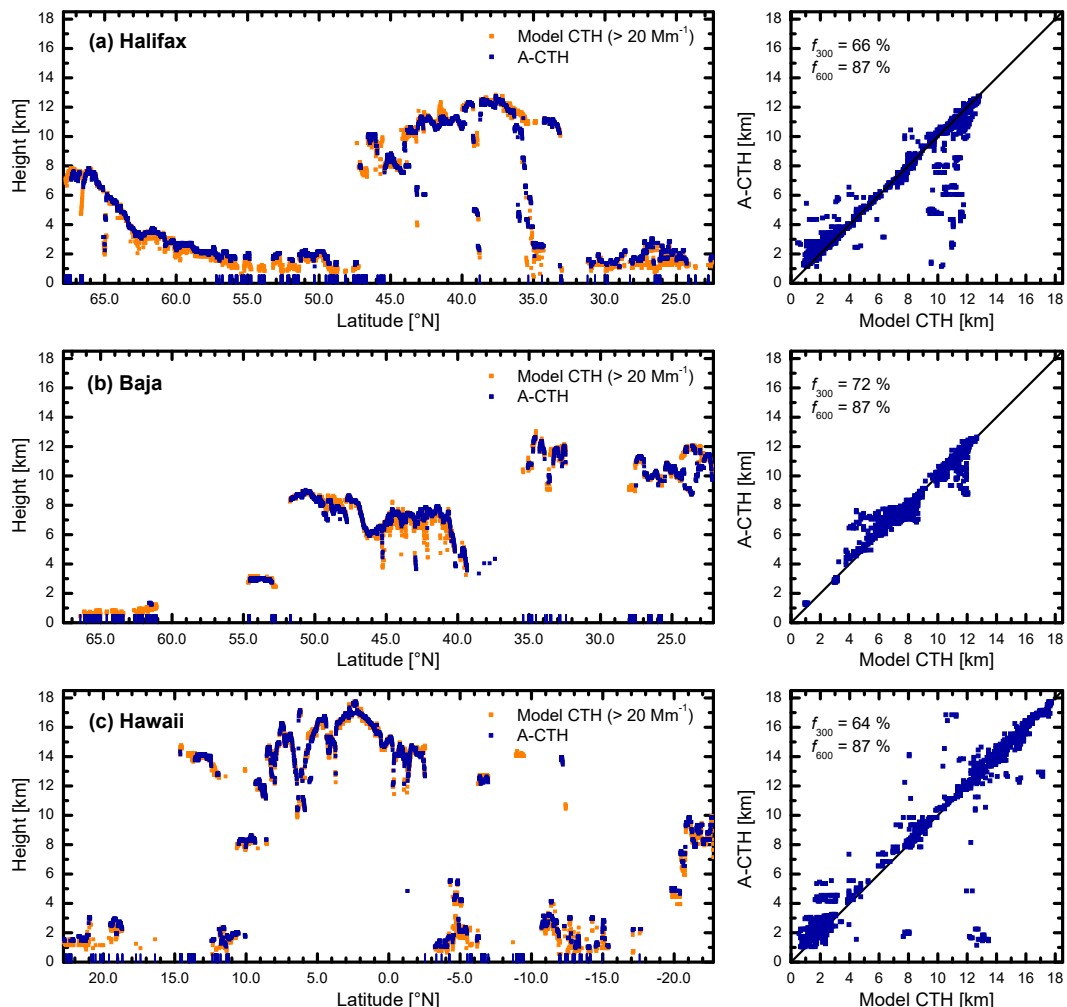

**Figure 8.** Comparison of the CTH derived with the A-CTH algorithm with the GEM model truth for (a) the *Halifax* scene, (b) the *Baja* scene, and (c) the *Hawaii* scene. The left panels show the direct comparison along the simulated satellite track and the right panels the respective scatter plots. The blue bars on top of the x-axis in the left panels indicate the pixels where a cloud was present in the model truth, but was not detected by the A-CTH algorithm. The numbers in the scatter plots show the percentage of agreement of data points within height intervals of $\pm 300$ and $\pm 600$ m, respectively. The extinction threshold for defining a cloud in the model output is set to 20 Mm$^{-1}$. The processing of all three scenes was performed with the same parameter settings given in Table 1.

The lower three panels of Fig. 9 show different quality indicators, namely the level of confidence for the top height of the lowermost aerosol layer ($C_T$, see Sect. 3.4.2), the level of consistency with the A-TC product ($X_{ALD} = (i, j)$, with $i$ in dark red and $j$ in blue, see Sect. 3.4.4), and the A-ALD quality status ($Q_{ALD}$, see Sect. 3.4.6). $C_T$ is mainly between 2 and 4, indicating a well pronounced layer top (WCT maximum well above the threshold value). Also the consistency with A-TC is very good and the quality status is mainly zero (highest quality). The spurious A-TC cloud detections ($X_{ALD} = (1, 0)$, $Q_{ALD} = 3$) south of






32° N are linked to the misinterpretation of noise peaks by the A-TC algorithm. Such effects could have been further reduced, if more aggressive configuration parameter tuning were carried out. However, it is not advisable to adapt the configuration parameters to a specific scene. Rather, optimization should take place at a later stage on the basis of a larger collection of real-world data.

In Fig. 10, several optical aerosol properties are presented for the *Halifax aerosol* scene. The two panels below the 2D cross section show different AOT products (see Sect. 3.4.3). For the present case with only one distinct aerosol layer, the AOT of the lowermost layer (layer 1) is equal to the sum of layer AOT for almost all pixels (black dots on orange symbols in the third panel). Accordingly, also the stratospheric AOT, indicated in gray in this panel, has values close to zero. The columnar AOT (blue in the second panel) is slightly higher than the sum of layer AOT, and the difference is indicated by the gray bars
called 'diffuse AOT'. It represents the contribution of aerosol that is not confined in distinct layers. Some of it is visible as a faint structure between 4 and 6 km height in the top panel. The mean optical properties of the lowermost layer are shown in the forth panel. This plot nicely demonstrates the ability of ATLID to provide layer-mean values of particle extinction and backscatter coefficients, lidar ratio, and linear depolarization ratio, which are the input for aerosol typing based on L2 products (Wandinger et al., 2022). The lowermost panel of Fig. 10 shows the columnar aerosol classification probabilities following
from this typing approach (see Sect. 3.4.5). Marine aerosol (blue) is correctly found as the major contributor to the AOT with a probability of >60 % for most of the pixels. Contributions of anthropogenic pollution (red) and mixtures with dust (dark yellow) are not completely ruled out. The respective probabilities are determined by the 2D Gaussian distribution functions that define the aerosol types in the lidar-ratio–depolarization-ratio phase space (see Wandinger et al., 2022, Fig. 9). The width of the distribution functions, and thus the overlap of neighboring types (here marine, pollution, and dusty mix), is configurable
in the A-TC algorithm and can thus be adjusted for real-world data if needed. The small contributions of ice and other aerosol types to the columnar classification probabilities are related to noisy data and respective misinterpretations as discussed above.

Finally, in Fig. 11, a comparison of the A-ALD columnar AOT with the model truth is presented for the three standard scenes, similar as for CTH in Fig. 8. The comparison is shown for all JSG pixels for which a solution is available in A-ALD (gray symbols) as well as for those pixels for which the quality status $Q_{\mathrm{ALD}}$ is zero (blue symbols, highest data quality). The
scatter plots in the right panels show that the agreement with the model truth increases when the quality status of the data is taken into account. However, deviations in AOT of >0.05 are still obtained for 10–40 % of the data points. As mentioned above, the standard scenes are dominated by clouds and thus aerosol retrievals are hampered. Adequate horizontal averaging for good extinction retrievals is not always possible, which leads to a higher uncertainty of the aerosol products. Moreover, the columnar AOT from the A-ALD product shows a positive bias of the order of 0.05–0.1 against the model truth in some regions.
The bias is mainly caused by contributions of optically very thin clouds that are not detected by the A-CTH algorithm and thus interpreted as aerosol in A-ALD. When $Q_{\mathrm{ALD}} = 0$ (blue symbols), also the A-TC product does not indicate any cloud in the profile. Most prominent in Fig. 11 is the range between 28° and 35° N in the *Baja* scene, where the bias is obviously caused by a very thin ice cloud at 10–13 km height with an optical depth of about 0.1 (see Qu et al., 2022b, supplement) and extinction values often below 20 Mm$^{-1}$ (i.e., no cloud indicated in the model truth in Fig. 8). Thin ice clouds not detected by A-CTH are





**Figure 9.** Illustration of the A-ALD product for the *Halifax aerosol* scene. In the top panel, the cross section of the simulated ATLID Mie co-polar signal is overlaid with bottom (gray) and top heights (black) of the lowermost aerosol layer. The surface return has been removed and horizontal smoothing over 11 JSG pixels has been applied for plotting the L1b signals. The second panel presents the comparison of the derived columnar AOT (blue) with the model truth (orange). The panels below show the level of confidence for the top height of the lowermost aerosol layer ($C_{\mathrm{T}}$), the level of consistency with the A-TC product ($X_{\mathrm{ALD}} = (i, j)$, with $i$ in dark red and $j$ in blue), and the A-ALD quality status ($Q_{\mathrm{ALD}}$). The processing was performed with the parameter settings given in Table 2.





**Figure 10.** As Fig. 9, but the panels below the 2D cross section show the columnar (blue) and diffuse AOT (gray); the lowermost-layer (orange), sum of layer (black), and stratospheric AOT (gray); the layer-mean optical properties for the lowermost aerosol layer; and the columnar aerosol classification probabilities.



**Table 2.** Configuration parameters used in the processing of the test scenes with the A-ALD algorithm.

| Configuration parameter | Symbol | Value |
|---|---|---|
| Dilation of the wavelet | $n_A$ | 12 |
| Troposphere partitioning parameter | $p_{trop}$ | 3 |
| WCT threshold, lower troposphere | $W_{A,1}$ | 0.05 |
| WCT threshold, upper troposphere | $W_{A,2}$ | 0.05 |
| WCT threshold, stratosphere below 20 km | $W_{A,3}$ | 0.05 |
| WCT threshold, stratosphere above 20 km | $W_{A,4}$ | 0.05 |
| SNR threshold, lower troposphere | $SNR_{A,1}$ | 1.5 |
| SNR threshold, upper troposphere | $SNR_{A,2}$ | 1.3 |
| SNR threshold, stratosphere below 20 km | $SNR_{A,3}$ | 1.3 |
| SNR threshold, stratosphere above 20 km | $SNR_{A,4}$ | 1.3 |
| Switch for internal layer structure | $s_{il}$ | 0 |
| Selection parameter for input data resolution | $s_{res}$ | 1 |
| Quality threshold for number of aerosol layers | $q_{A,nal}$ | 2 |
| Quality threshold for relative backscatter error | $q_{A,bsc}$ | 0.1 |
| Quality threshold for consistency with A-TC | $q_{A,con}$ | 8 |

also present near the surface between 61° and 66° N in the *Baja* scene, but they are screened by the comparison with A-TC ($Q_{ALD} = 3$).

In general, from the A-ALD algorithm tests, it can be concluded that combining information from A-ALD and A-TC, together with proper settings of the configuration parameters in both algorithms, helps optimize cloud–aerosol discrimination and thus the quality of the ATLID aerosol products. As for A-CTH, careful testing and validation of the A-ALD algorithm with

real-world data for a wide range of scenarios after the EarthCARE launch is important.

## 5  Conclusion and outlook

The A-LAY processor has been developed to generate cloud-top-height and aerosol-layer information from ATLID L1 and L2 profile data. The A-CTH and A-ALD products serve as input for the synergistic AM-COL processor, which combines data from ATLID and MSI to extend information from the ATLID track to the MSI swath (Haarig et al., 2023). Therefore, all

data are provided on the EarthCARE Joint Standard Grid. A wavelet-covariance-transform method with flexible thresholds is used to determine cloud and aerosol layer boundaries from ATLID Mie co-polar signals with different horizontal resolution. Appropriate threshold settings allow the detection of optically thick clouds at the native JSG resolution (approximately 1 km horizontal, 100 m vertical). For thin clouds and aerosol layers horizontal averaging over 11 JSG pixels is applied. Next to geometric layer boundaries, the products contain further information on cloud and aerosol layers. A simplified classification





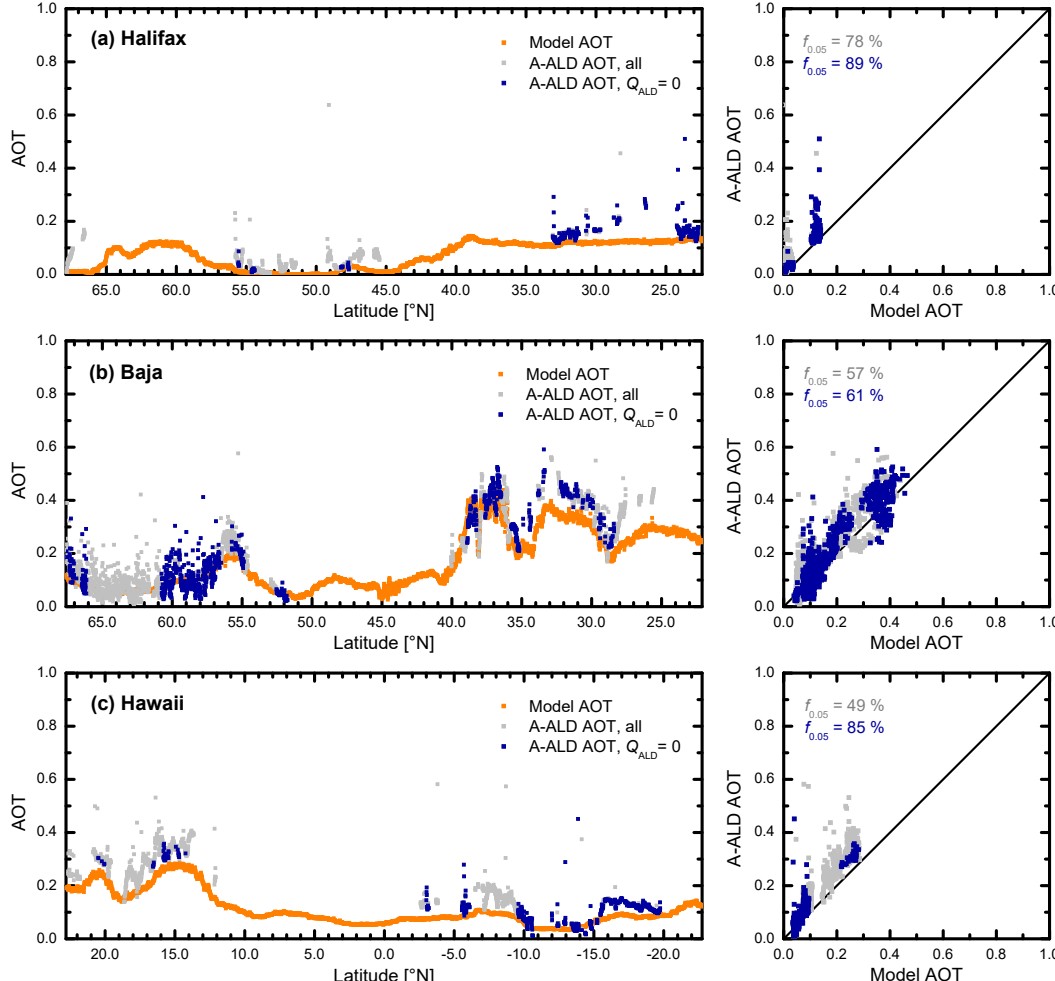

**Figure 11.** Comparison of the A-ALD columnar AOT with the model truth for (a) the *Halifax* scene, (b) the *Baja* scene, and (c) the *Hawaii* scene. The left panels show the direct comparison along the simulated satellite track and the right panels the respective scatter plots. Gray symbols show all pixels for which a columnar AOT is provided in the A-ALD product, while blue symbols indicate those pixels for which the quality status $Q_{ALD}$ is zero. The numbers in the scatter plots stand for the percentage of agreement of data points within an AOT interval of $\pm 0.05$. The processing of all three scenes was performed with the same parameter settings given in Table 2.

of the uppermost cloud, including multi-layer information, is provided as input for the synergistic AM-CTH algorithm. For aerosol layers, layer-mean optical data are calculated from extinction, backscatter, lidar-ratio, and depolarization-ratio profiles at 355 nm taken from the A-EBD product. Furthermore, the AOT of each layer, the stratospheric AOT, and the columnar AOT are stored in the A-ALD product. Later in the processing chain, the columnar AOT at 355 nm is combined with the AOT at longer wavelengths from MSI to derive synergistic Ångström exponents. Both aerosol and cloud parameters are compared with



results from the A-TC product and respective consistency parameters are calculated. In addition, several quality criteria are applied and the results are stored in the products as well.

The atmospheric test scenes, which were created for developing, testing, and evaluating all EarthCARE processors (Qu et al., 2022b; Donovan et al., 2023b; van Zadelhoff et al., 2022), have been used to demonstrate the functionality of the A-CTH and A-ALD algorithms. It could be shown that the algorithms perform as expected, in particular regarding the detection

of cloud and aerosol layer boundaries. The flexible configuration parameters (e.g., dilation of the wavelet function, threshold values for WCT and SNR) help adjust the algorithms to the actual observational and instrumental conditions. In principle, specific settings, e.g., for day/night conditions or certain geographic locations, are possible. Such kind of fine-tuning requires the analysis of a larger amount of real-world data when the mission is in space. Validation of the products by independent ground-based and airborne measurements is a crucial task in this context. Thus, joint efforts of algorithm developers and

validation teams are highly desirable, as for all EarthCARE products. The algorithm developments will continue until the launch of EarthCARE, with a specific focus on tests for stratospheric aerosols and clouds. Further algorithm improvements are planned throughout the course of the mission, taking lessons learned into account.

*Data availability.* The EarthCARE Level-2 demonstration products from simulated scenes, including the A-CTH and A-ALD products discussed in this paper, are available from https://doi.org/10.5281/zenodo.7311704 (van Zadelhoff et al., 2022).

*Author contributions.* UW has developed the described algorithms and drafted the manuscript. HB contributed the basics of the WCT methodology. MH has performed numerous tests and made various improvements to the algorithms. DD and GvZ provided the input data from the A-PRO processor. All authors were involved in the discussion of algorithm developments and tests and contributed material and/or text to the manuscript.

*Competing interests.* UW is member of the editorial board of Atmospheric Measurement Techniques and co-editor of the Special Issue to
which this paper contributes. The peer-review process was guided by an independent editor. The authors have no other competing interests to declare.

*Acknowledgements.* This work has been funded by ESA grants 4000112018/14/NL/CT (APRIL) and 4000134661/21/NL/AD (CARDINAL). We thank Stefan Horn and Florian Schneider who helped with the technical implementation of the code. We are grateful to Tobias Wehr and Michael Eisinger for their continuous support over many years, and we thank the EarthCARE developer team for valuable discus-
sions in various meetings.

The work on this manuscript was overshadowed by the sudden and unexpected passing of ESA's EarthCARE Mission Scientist Tobias Wehr. He is greatly missed. His tireless commitment to the mission and the science community around EarthCARE will not be forgotten.



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
