# Peer review of "Cloud top heights and aerosol layer properties from EarthCARE lidar observations: the A-CTH and A-ALD products"

_EGUsphere, 2023_

## Author Comment (AC1)

**Reply to the comments**

We thank the three anonymous referees for their kind evaluation of the manuscript. Referee #1 did not request any changes. The comments of Referees #2 and #3 helped us to add necessary information and clarifications. Please find our answers (blue text) to the comments (black text) below. Please note that we have numbered the specific comments of Referee #2 so that we can refer to them. Respective changes to the manuscript are indicated in blue in the version attached to this reply. Line numbers in our answers refer to this manuscript version. Tracked changes do not include minor corrections of spelling/grammar not related to the referee comments.

**Referee #2**

The paper describes the EarthCARE cloud top height and aerosol layer products (A-CTH, A-ALD). The paper is very well-written, logically organized, and the methods employed are sound. However, the motivation for this study was never quite clearly laid out in the manuscript. The authors also do mention a desire to maintain the heritage of the CALIPSO layer products, which is worthwhile. But why then develop an additional independent data products with independent methods when there already exists cloud and aerosol profiles products from ATLID (i.e. A-FM/APRO)? Layer products are indeed useful when it is more convenient to use layer data rather than more detailed profile data, but it seems more desirable to make the information provided by the layer product consistent with the profile product. That is: it would seem more prudent to derive the layer product from directly form the information given in the profile products rather developing a different algorithm.

While a flag is developed to alert a data user to such disagreement between these layers data products and their profile counterparts, how does one then decide which information is more reliable to use for their purpose? I'm concerned about the potential pitfalls of providing data users with 2 independent descriptions of cloud+aerosol locations and properties.

We understand the concern. The reason lies in the history of the algorithm developments. ESA started parallel projects with independent developer teams about 15 years ago. Since no L2 products were existing at that time, every group had to start with L1 data. Therefore, the layer detection algorithm was developed to retrieve cloud top heights and aerosol layer boundaries as prerequisites for the synergistic ATLID-MSI algorithms. ATLID profile products (as well as other products) were developed in parallel. Later on, the activities were successively merged, and a single processing chain was established. Therein, the A-LAY processor was placed behind the A-PRO processor. In this way, it was possible to extend the algorithm and use L2 profile data to calculate, e.g., layer-mean optical properties. In the end, we introduced the consistency checks to provide a tool for dealing with the two different approaches.

At first glance, the use of different approaches may look like a pitfall. However, from our experience, it is very helpful to have more than one method at hand to derive certain information (which by the way is true for several EarthCARE products). Consistency checks help verifying the results and show uncertainties and limitations of the retrievals. Users of spaceborne products should always critically check whether data and data accuracy serve their purpose. For applications that use lidar data in combination with passive remote sensing, the ATLID layer products are well suited since they provide quite robust and easy-to-access information on the presence of clouds and the vertical location of distinct aerosol layers. Other studies, e.g., on aerosol-cloud interactions, will need more detailed information on the vertical structure of the atmosphere and a sophisticated height-resolved aerosol-cloud discrimination and may thus better rely on the ATLID profile products.

We have reformulated a part of the introduction to provide some more description regarding the different approaches (see lines 50–70), and we have added a paragraph regarding the potential use of the products in the conclusions (see lines 580-586).

Other comments:

1. line 108: why clear conditions only? One of strengths of lidars being able to provide aerosol information in the presence of clouds.

As mentioned, the ATLID layer products have been developed as prerequisites for the synergistic ATLID-MSI algorithms. The distinction between clear and cloudy pixels allows a one-to-one allocation of the A-CTH and A-ALD products to the MSI cloud mask and the subsequent combination with either the M-COP cloud product or the M-AOT aerosol product. Therefore, we skipped the search for aerosol layers in the presence of clouds. However, we agree that providing aerosol information in the presence of clouds is a strength of lidars. Users can get this information from the ATLID profile products (A-TC, A-EBD, A-AER), but it could be considered in the layer products as well by implementing an additional search loop and an additional classification parameter (aerosol above/below cloud). This functionality can be foreseen for future algorithm improvements.

We have added some explanations in Sect. 2.2 (see lines 121-123).

Please also see the answer to Comment #6 of Referee #3.

2. This layer information appears to be as a needed precursor to the MSI cloud and aerosol retrievals. But, in a similar vain to my comments above, why couldn't the profile version of the products be used for this purpose?

As explained above, profile products have not been available when the developments were started, and it was required by the Agency to use independent approaches (changes made as indicated above).

3. A ~10km average (i.e. 11 JSG pixels) seem insufficient to detect optically thinner layers as evident by the evident application of the algorithm to the Halifax aerosol scene. Why is 11 pixels chosen as the coarsest resolution? Are these undetected thin clouds/aerosols expected to impact the MSI retrievals?

The chosen resolution is based on the mission requirements which stipulate a scene reconstruction (for aerosol, clouds, precipitation) based on a 10 km x 10 km footprint for radiation closure assessments. Accordingly, products are required with 10 km horizontal resolution (see Wehr et al., 2023 and the Mission Requirement Document, https://doi.org/10.5270/esa.earthcare-mrd.2006).

In principle, the resolution can be changed in the A-LAY processor if needed. However, for the purpose of chaining and combining the ATLID and MSI products, a common resolution is required, and therefore a free configuration of the averaging is not considered. If the validation of the algorithm after launch shows the need for changing the resolution, this will be possible. From the tests, we know that the 11-pixel average provides already a better cloud screening than the MSI cloud mask, thus the MSI retrievals are improved with the synergistic approach (see the paper on the AM-CTH and AM-ACD products by Haarig et al., 2023). In general, the uncertainty of the MSI products is larger than the influence of very thin cloud/aerosol layer not detected with A-LAY.

We have added an explanation in the beginning of Sect. 2 (see lines 99-101).

Please also see the answer to Comment #5 of Referee #3.

4. For the test scenes, it be would be good to summarize the level of consistency between the (e.g. statistics of the level of consistency flag in Figure 7).

Thank you for the recommendation. We have added statistics for $X_{CTH}$ in the discussion of Fig. 7 (see lines 467-471) and also provide statistics for $Q_{CTH}$ in the discussion of Fig. 8 (see lines 494-497) and for $Q_{ALD}$ in the discussion of Fig. 9 (see lines 522-524) and Fig. 11 (see lines 547-548 and 559).

5. How sensitive are the results using the test scenes to the parameters chosen in Tables 1 and 2? Relatedly, have the authors thought about the process for optimizing these once EarthCARE is collecting real data?

Of course, we have played with the parameters and selected them such that optimum results are obtained. The optimum was relatively easy to find by maximizing the number of correctly detected cloud and aerosol-layer boundaries, and it was not necessary to change the parameters afterwards for the different test scenes. Thus, the sensitivity is high enough for an optimization, but not so high that it leads to an instability of the algorithm when it is applied to different scenes.

It is clear that the parameters need to be adjusted for the real data by considering the actual signal-to-noise ratio (which will change depending on location/time of observation and over the lifetime of the mission). Optimization will be done during the commissioning phase, e.g., by analyzing probability distributions of detected cloud and aerosol-layer boundaries for selected scenes, different frames (corresponding to different pieces of the orbit) etc. The results will be validated with independent observations from ground (lidar and radar network observations), from aircraft (underflights with lidar and radar instruments during campaigns), and cross-satellite comparisons if available. There is also room for algorithm improvements if needed, for instance when the false detection rate (e.g., due to noise peaks) turns out to be too high.

The respective discussion in Sect. 4.1, 4.2, and 5 has been extended (see lines 447-448, 474, 488-491, 512, 593-595).

Please also see the answer to Comment #3 of Referee #3.

**Referee #3**

This manuscript provides an introduction to the products of cloud top height and aerosol layer properties derived from EarthCARE lidar. Retrieval algorithms are discussed, and validation is carried out using model simulations. This study is of significant importance to the EarthCARE satellite mission and deserves documentation as it serves as a valuable resource for future data users, facilitating a better understanding of the data products and providing useful guidance. The manuscript is generally well-written, the methodology is sound, and the conclusions are solid. I recommend a minor revision prior to publication. Please find my detailed comments below:

1. While the manuscript provides a good discussion of the algorithm, it is not an Algorithm Theoretical Basis Document (ATBD). The objective of this paper could be better tied to the broader needs of the community. I believe it would be worthwhile to highlight the advantages and benefits of the EarthCARE lidar and the algorithm in the manuscript. Additionally, the paper could benefit from discussing what unique aspects of the products future data users can anticipate.

Thank you for the recommendation. We have extended the explanations in the abstract, introduction, and conclusion accordingly (see lines 2-6, 73-74, 82, 580-586). We would like to emphasize that there is an agreed guideline for all papers contributing to the Special Issue on EarthCARE Level 2 algorithms and data products not to repeat general descriptions of the mission, its instruments, goals, requirements etc. but to refer to the respective papers of the Special Issue, particularly the overview papers, instead. Therefore, we focus only on the algorithm and products described in this paper.

2. The manuscript contains numerous acronyms, which can become confusing. A table summarizing all acronyms in a single place would be appreciated.

We agree that dealing with all the abbreviations of such a big mission like EarthCARE is challenging. Most of the acronyms used in this paper refer to the algorithms and data products of the EarthCARE mission as a whole. They follow the definitions agreed by the Agencies and are commonly used throughout the Special Issue and in all EarthCARE-related documents. An overview of the entire production chain with all algorithm and data product names is given in the paper by Eisinger et al. (in preparation), and a brief summary is also provided in Wehr et al. (2023). Therefore, we do not want to come up with a specific solution (table or glossary) just for this single paper. We follow the common rule of introducing an acronym the first time it appears in the text, and we refer to the overview papers in several places.

3. I would appreciate a discussion on how the accuracy of the retrieval depends on the chosen configuration parameters.

We have added some discussion in Sect. 4.1, 4.2, and 5 (see lines 447-448, 474, 488-491, 512, 593-595). Because of the limited number of scenarios (e.g., cases with dense aerosol layers are missing in the test scenes), a deeper analysis of the dependency of results on the parameter settings was not possible. So far, the parameters are chosen "to the best of our knowledge", based on manual settings of the parameters and visual inspection of the results. An in-depth evaluation and optimization of the parameter settings will be performed with real-world data after the launch of EarthCARE.

Please also see the answer to Comment #5 of Referee #2.

4.  Given that modeled cases are utilized to validate the algorithm, I would like to suggest performing a statistical performance analysis using a larger set of model outputs, as opposed to a limited number of cases.

We have used all available model cases for the analysis. The generation of the three test scenes has been a very huge and time-consuming effort of the developer team (see Qu et al., 2022; Donovan et al., 2023). The scenes have been generated to demonstrate the performance of all EarthCARE algorithms and to test the entire processing chain. Although this has been a very valuable exercise, it is not considered to be an algorithm validation, even not when more scenes would be available, for several reasons. First of all, the resolutions of the NWP (GEM) and aerosol (CAMS) models are limited and provide only a bulk description of the atmosphere (see Qu et al., 2022). Secondly, ECSIM is run on top of the atmospheric models and assigns only a limited number of scattering characteristics to the simulated targets. Finally, a certain instrument performance is assumed to generate realistic, noisy signals. Thus, we put all our expectations on atmospheric and instrument properties into the simulation and check the retrievals against the expectations and requirements. Such an end-to-end exercise is well suited for a first verification of the algorithms, but it is not a validation.

Validation of EarthCARE algorithms and data products is a post-launch effort based on intercomparisons against independent external references. For this purpose, ESA has initiated an international Cal/Val programme involving ground-based, airborne, cross-satellite, and modelling approaches. The commissioning phase (first six months after launch) is specifically dedicated to the verification, calibration, and validation of instruments, algorithms, and products. The algorithm developers will use this time for an in-depth investigation of the EarthCARE products and propose algorithm improvements if needed. Validation will then continue over the lifetime of the mission.

5.  Line 94: Can you explain the reason for choosing 11-pixel for this algorithm?

The chosen resolution is based on the mission requirements which stipulate a scene reconstruction (for aerosol, clouds, precipitation) based on a 10 km x 10 km footprint for radiation closure assessments. Accordingly, products are required with 10 km horizontal resolution (see Wehr et al., 2023 and the Mission Requirement Document, MRD, https://doi.org/10.5270/esa.earthcare-mrd.2006). A gliding average over 11 JSG pixels comes closest to this requirement. Note that the JSG is determined by the radar footprint and is somewhat irregular due to the radar cycle. Only in case of missing radar measurements, the pixel length is set to exactly 1000 m. One JSG pixel may contain 3 or 4 lidar profiles (with 285 m distance for a 2-shot on-board accumulation).

We have added an explanation in the beginning of Sect. 2 (see lines 99-101).

Please also see the answer to Comment #3 of Referee #2.

6.  Line 108: Why only clear sky? There are two crucial scenarios that need to study the aerosol radiative effects, namely aerosols above low-level water clouds and aerosols beneath thin cirrus. These datasets are currently missing from existing satellite products.

As already mentioned, the ATLID layer products have been developed as prerequisites for the synergistic ATLID-MSI algorithms. The distinction between clear and cloudy pixels allows a one-to-one allocation of the A-CTH and A-ALD products to the MSI cloud mask and the subsequent combination with either the M-COP cloud product or the M-AOT aerosol product. Therefore, we skipped the search for aerosol layers in the presence of clouds. However, we agree that aerosol layers above low water clouds and below thin cirrus are crucial scenarios that should be covered. In principle, users can get this information from the ATLID profile products (A-TC, A-EBD, A-AER). We could consider it for the layer products as well by implementing an additional search loop and an additional classification parameter (aerosol above/below cloud). Thus, yes, we agree, and we have it mind for future algorithm improvements.

We have added some explanations in Sect. 2.2 (see lines 121-123).

Please also see the answer to Comment #1 of Referee #2.

7. Line 111: The assumption being made here is that the aerosol is vertically well-mixed within the boundary layer. Could you elaborate on the uncertainty associated with this assumption?

Actually, there is no such assumption made. As for any other layer, there is only the condition that the aerosol load is high enough to cause a signal-to-noise ratio above the threshold value. The algorithm even provides an option (via a configuration parameter) to search for internal layer boundaries, which do not occur when the layer is well mixed.

We have added a small clarification (see line 127).

8. Line 293-295: In cases where QA equals 2 and 3, which data should a data user use?

As mentioned above, each user should critically check under which conditions data are useful for a certain purpose. Highest confidence is given (for all EarthCARE products) when the quality status of a product is zero. In all other cases, the reasons should be checked, and the decision whether data can be used at all should be made based on the application. For example, when users need a cloud-conservative result, they should not use the data when $Q_{CTH}$ = 3. When a high confidence in the top height is required, neither data with $Q_{CTH}$ = 2 nor with $Q_{CTH}$ = 3 should be used.

9. Line 371-372: Same question as above.

Again, we can only recommend to use the data according to the purpose. For instance, if a clear-conservative result is needed, data with $X_{ALD}$ = (1, 0) and $Q_{ALD}$ = 3 should be skipped.

[revised manuscript text omitted]